# Influence of Cotton Pre-Treatment on Dyeing with Onion and Pomegranate Peel Extracts

**DOI:** 10.3390/molecules27144547

**Published:** 2022-07-16

**Authors:** Lea Botteri, Anja Miljković, Martinia Ira Glogar

**Affiliations:** Faculty of Textile Technology, University of Zagreb, 10000 Zagreb, Croatia; lea.botteri@ttf.hr (L.B.); amiljkovi@ttf.hr (A.M.)

**Keywords:** cotton yarn, scouring, chemical bleaching, mercerization, natural dyes

## Abstract

In this paper the possibility of applying natural dyes on cellulose fibres were researched with respect to the impact of cotton material pre-treatment (scouring, chemical bleaching, mercerization and mordanting), using renewable sources of natural dyes (waste as a source). As mordants, metal salts of copper, aluminium and ferrum were used, and the influence on colour change as well as on fastness properties were analysed. The natural dyes were extracted from onion peel (*Allium cepa* L.) and pomegranate peel *(Punica granatum* L.). In spectrophotometric analysis performed of the plant extracts, the onion extract has peaks at 400 and 500 nm, resulting in red-orange colourations and the pomegranate extract shows a maximum at 400 nm, i.e., in the yellow region, which is characteristic of punicalin. Results show significant influence of cotton pre-treatments on colour appearance and fastness properties, caused by pre-treatments affecting the properties and structure of the cotton itself. The positive effect of mercerization on dye absorption and bonding is confirmed. For wash and light fastness properties, more satisfactory results have been obtained for yarns dyed with pomegranate peel natural dye, and the key importance of mordants for fastness properties has been confirmed.

## 1. Introduction

Recent decades have witnessed scientific research efforts aimed at revitalizing and commercializing the use of natural dyes in textile dyeing. There are many positive aspects of natural dyes, which justify this work and effort: from the positive colour characteristics of which the colour hues achieved by natural dyes are always in a harmonious and balanced relationship that positively affects the user, to their proven antibacterial properties, protective properties form harmful UV radiation and environmental sustainability and non-toxicity [1,2,3,4,5]. However, the application of natural dyes in commercial production has certain obstacles. Natural dyes many times lack in uniformity and reproducibility of colour. In addition, the problem is their availability in bulk quantity. There is no sufficient information on standardization of natural dyes application methods, as well as often-questionable colourfastness properties [6]. The application of natural dyes on cotton attracts special attention due to the popularity of cotton fibre, but this is where most of the problems are encountered, due to the natural affinity of most natural dyes for not cellulose but protein fibres. Ferreira conducted a comprehensive study of flavonoid dyes, among which emphasizes research into the chemical structure of onion peel dyes. The results provided important structural information on previously unidentified flavonoids in natural yellow dye extracts. The study contributes greatly to the understanding of the responsibility of individual compounds contained in the dye structure of onion peel, to achieve a certain colour hue, but also for the fastness properties, mostly light fastness [7]. Ferreira et al. also published a comprehensive study of the chemical constituents of the main classes of natural dyes, pointing out the importance of the possibilities provided by modern experiment methodology in establishing chemical correlations with historical records, and the importance of understanding the sensitivity of individual compounds to photo-degradation, thus allowing a more precise definition of the conditions of storage of historical textiles [8]. They also researched the identification and photochemical degradation of flavonoid dyes using photo-oxidation products of quercetin and morin as a marker for the characterization of natural yellow dyes in ancient textiles [9].

Papers dealing with topics similar to this paper confirms the extensive research work invested in studying the mechanisms of dyeing cellulosic materials with natural dyes and their binding to cellulose, as well as the influence of mordant, not only on significant colour changes, but also in the context of their key role in achieving satisfactory colour fastness. Iqbal and Liaqat researched the possibility of applying the natural dye extracted from pomegranate peel on cotton fabric and achieved a satisfactory results of colour depth, intensity and wash fastness properties by using the oxalic acid as mordant [10]. The antibacterial, biocidal and deodorizing properties of the natural dye obtained from pomegranate peel and applied at cotton have been also confirmed [11,12]. Davulcu et al. [13] investigated the possibility of applying natural dyes of thyme and pomegranate peel on cellulose fabrics, analysing the influence of mordant on the colour characteristics as well as on washing, rubbing, perspiration and light fastness properties. Using potassium aluminium sulfate, copper (II) sulfate, iron (II) sulfate and tin (II) chloride as mordant, interesting conclusions were reached about minimal or even no effect of mordant on wash fastness of samples dyed with thyme dye. For pomegranate peel dyed samples, also the premordanting process did not enhance the washing, rubbing or perspiration fastness properties, which was satisfactory already on non-mordanted samples. As for the light fastness, they confirmed the key role of mordants in achieving the satisfactory results. They also confirm the antibacterial properties of the samples dyed without prior mordanting. The similar work has been conducted by Kulkarni et al. [14] with the proviso that they analysed the impact of pre-, during dyeing- and post-mordanting. Satyanarayana and Remesh Chandra [15] researched the possibility of applying pomegranate peel dyes on cotton fabrics with prior scouring of cotton fabric and mordanting with copper sulphate and ferrous sulphate. They obtained moderate rub, good wash and very good light fastness of dyed samples. Adeel et al. [16] focused on optimizing the dyeing process with natural dye of pomegranate peel by monitoring the influence of dyeing temperature and time and salt concentration as an additive in the dyeing bath. In their conclusion, they also confirmed that despite ongoing research, there are still many open areas where a solution needs to be found. Silva et al. [17] investigated the application of onion skin dye on cotton fabric with the possibility of using chitosan as the replacement for electrolyte in dyeing process. As the results they obtained acceptable levels of fastness and antimicrobial as well as finding that chitosan contributed to an increase in UPF value as the measure of protective properties against harmful UV exposure. Such research confirms great interest in the field of application of natural dyes on textiles as well as finding the possibility of returning to natural dyes, not only in localized production of small series but also in wider, even industrial production.

Regarding the application of natural dyes in modern production, the important aspect is the sustainability of dye sources. Cheap by-products from agriculture and forestry can be used to obtain natural plant dyes, e.g., wood bark from the timber industry, waste from the food and beverage industry such as pressed berries, distillation sludge and other residues. The source of natural dyes can also be, for example, wastewater from olive mills, which is a by-product of olive oil extraction and causes major environmental problems in Mediterranean countries. An intensive use of industrial wastes as renewable raw materials for the production of natural dyes would increase the economic use of waste materials, contribute to the protection of the environment and reduce the use of fossil fuels [18,19,20,21,22,23,24]. This approach to the recovery of waste as a source of raw materials fits perfectly into the zero-waste concept, which implies integrated recycling, material reuse, resource optimization, waste reduction and deconstruction.

In this paper, the complex issue of applying natural dyes on cellulosic materials is addressed from two aspects: first, the possibility of applying natural dyes on cellulose fibres with respect to their natural affinity for protein fibres and the impact of cotton material pre-treatment (scouring, chemical bleaching, mercerization and mordanting) and second, the possibility of using renewable sources of natural dyes (waste as a source). The plants used, onion and pomegranate, traditionally grow in Croatia. Extracts obtained from the waste of these plants were used in dyeing cotton yarn. Raw, scoured, chemically bleached, mercerized and mordanted cotton yarns were dyed with natural dyes obtained from onion peel (*Allium cepa* L.) and pomegranate peel (*Punica granatum* L.). Cotton yarns dyed with natural dyes has added value not only for environmental and economic reasons but also from a social and psychological aspect. This research contributes to the complex problem of dyeing cellulose fibres with natural dyes, but also to the circular economy.

## 2. Methods

The 100% raw cotton yarns^®^, Nm 16/2, Z-15 (Unitas, Zagreb, Croatia) were scoured (S), chemically bleached (CB) and mercerized (RM, RM4). After the pre-treatments, the cotton yarns were dyed with onion peel (*Allium cepa* L.) and pomegranate peel (*Punica granatum* L.). The labels and treatments are shown in Table 1.

### 2.1. Pre-Treatments of Cotton Yarns

Scouring was performed in sodium hydroxide solution with the addition of wetting. The natural pigments remaining after scouring were removed by chemical bleaching with hydrogen peroxide in an alkaline medium with the addition of organic and inorganic stabilizers, sequestering agent and antifoaming agent. Slack mercerization was performed in a laboratory beaker containing 24% sodium hydroxide solution with the addition of a wetting agent. Cotton yarns for mercerization with tenacity were placed in the frame of the mercerizing machine and immersed in 24% sodium hydroxide solution with the addition of a wetting agent. The cotton yarns were twisted in both directions at 4% of tenacity. The pre-treatment procedures are listed in Table 2.

### 2.2. Dye Extraction

Onion peel and pomegranate peel were used for dye extraction (Figure 1). Onion peel (*Allium cepa* L.) is rich in flavonoids, the most abundant of which is quercetin (2-(3,4-dihydroxyphenyl)-3,5,7-trihydroxy-4H-chromen-4-one), followed by quercetin glucoside and kaemfoerol. Due to their chemical structure, flavonoids are the most abundant plant dyes of yellow hue. Pomegranate peel (pomegranate, *Punica granatum* L.) contains 28% tannins. The hydrolysate may be gallic acid, ellagic acid and flavogalol. The tannins are punicalagin (2,3-(S)-hexahydroxydiphenoyl-4,6-(S,S)-galagyl-D-glucose; α-punicalagin; β-punicalagin) and puniclin (4,6-(S,S)-galagyl-D-glucose), which are responsible for the yellow colour [25,26].

Dye extraction was performed in soft water containing 100 g/L onion peel and pomegranate peel. The extraction was performed in a 1:40 bath ratio (considering the mass of the plant) at 100 °C for 60 min. The bath was then allowed to cool for 12 h and the extract was decanted.

Spectrophotometric analysis of the plant extracts was performed on a Cary 50 Solascreen, Varian absorption spectrophotometer in the ultraviolet (250–400 nm) and visible parts of the spectrum (400–700 nm).

### 2.3. Mordanting of the Textile Material with Metal Salts

Onion peel and pomegranate peel belong to the group of acid mordant dyes due to their dyeing properties and bind to fabric by forming a complex with metal salts, called mordants. For this reason, some samples were treated with potassium aluminium sulphate dodecahydrate KAl(SO_4_)_2_·12H_2_O, copper (II) sulphate pentahydrate CuSO_4_·5H_2_O and ferrous (II) sulphate heptahydrate FeSO_4_·7H_2_O; (Kemika, Zagreb, Croatia) before the dyeing process [27].

Pre-treatment of yarns with metal salts was performed with 5% mordants (based on the mass of the material) in a bath ratio of 1:30 in a Polycolor Mathis apparatus at 50 °C for 30 min. After pre-treatment with metal salts, the cotton yarns were rinsed with cold water.

### 2.4. Dyeing with Natural Dyes

Dyeing with natural dyes using onion peel and pomegranate peel of pre-treated cotton yarns was performed with a bath ratio of 1:30 in a Polycolor Mathis apparatus at 60 °C for 60 min. After dyeing, the cotton yarns were rinsed with cold water. Since these natural dyes are from the group of acid-mordant dyes, dyeing was performed at a pH of 4 adjusted with 20% acetic acid (Kemika, Zagreb).

### 2.5. Colour Analysis in the CIEL*a*b* System

Colour characteristics were measured using a remission spectrophotometer, Datacolor 850, measuring geometry d/8°, D65, measuring aperture of 9 mm. The whiteness (W_CIE_) of undyed cotton yarns were performed according to ISO 105-J02:1997 Textiles—Tests for colour fastness—Part J02: Instrumental assessment of relative whiteness. The coordinates used to determine colour values are “L*” for lightness, “a*” for redness (positive value) and greenness (negative value), “b*” for yellowness (positive value) and blueness (negative value), “C*” for chroma and “h” for hue angle in the range of 0° to 360° of undyed and dyed cotton yarns were determined according to ISO 105-J01:1997 Textiles—Tests for colour fastness—Part J01: General principles for measurement of surface colour.

All results were measured on samples by repeating the measurement procedure at random locations on the samples. Thus, the colour measurements were made using the Datacolor Tools computer program and “Measuring until tolerance” command, which means that at least 10 measurements must be made, and the results are accepted only if the total colour difference between each measurement is less than 0.1 (dE* < 0.1).

### 2.6. Wash Fastness

Wash fastness of cotton yarns were tested in a laboratory apparatus for wet and dyeing processes Polycolor, Mathis. The test was performed according to standard ISO 105-C06:2010 (A2S) Textiles—Tests for colour fastness—Part C06: Colour fastness to domestic and commercial laundering, using 2 g/L of standard detergent (James Heal ECE A, without optical brighteners and without phosphates), with a bath ratio of 1:20, temperature of 40 °C, time of 30 min. The results of wash fastness are given as numerical values of total colour difference calculated according to CIE76 equation, as well as in grey scale grades, obtained by comparing unwashed samples with samples that were washed after the 1st and 5th washing cycle.

### 2.7. Lightfastness

Lightfastness testing of the samples were performed on Xenotest 440 ((SDL Atlas, Rock Hill, SC, USA). Xenotest 440 is used for laboratory simulation of external weather influences on the stability and durability of textile and other materials. Analysis was evaluated according to the modified ISO 105-B02 and 13 B04 test methods using Xenotest 440. Test conditions simulated in this research were: Total light time: 41:10 h, Radiant exposure: 6226 kJ/m^2^, Irradiance control: 300–400 nm, Filter system: B04, E: 42 W/m^2^ (±2 W/m^2^), CHT: 32 °C (±3 °C), BST: 47 °C (±8 °C), RH: 40% (±8%), no spray, fan speed: 2000 rpm. Using the same equation as for the wash fastness, the lightfastness properties was also evaluated by calculating total colour difference values (DE), as well as in blue scale grades, obtained by comparing samples before and after exposure.

## 3. Results and Discussion

In this paper the influence of pre-treatment of cotton material on dyeing with natural dyes is presented. Pre-treatments of scouring, chemical bleaching and mercerization were performed. After the process of scouring, a part of the samples was chemically bleached and a part was left only scoured, while the mercerization process was carried out directly on the raw samples without prior scouring and bleaching.

In the first step, the spectrophotometric analyses of whiteness (W), yellowness (YI) and lightness (L*) have been performed on undyed samples, regarding the different pre-treatments. The objective results of whiteness, yellowness and lightness of undyed treated samples are shown in Table 1, while the photographs of samples are given in Table 3.

In addition to cellulose, cotton fibre contains impurities that give cotton its hydrophobicity and thus prevent satisfactory treatments of cotton material. The scouring has been performed in order to remove the hydrophobic impurities from the primary wall (e.g., pectin, proteins, and organic acids) and cuticle (waxes and fats) from cotton fibre [25,28,29,30]. The spectrophotometric measurement of whiteness of raw material showed rather low value (W_CIE_ = −17.00), which is to be expected since raw cotton has impurities that give it a yellowish colour. Scouring removes all impurities except pigments, and whiteness increases (W_CIE_ = 40.10), while the highest whiteness is obtained with chemical bleaching (W_CIE_ = 79.70), causing the natural pigments to be removed, resulting in a significant increase in whiteness. In this way, the results of dyeing become accurate. Due to the protoplasmic residues of the protein and flavone pigments of the cotton flowers, it has its natural greyish colour. By chemical bleaching, the cotton obtains a permanently white surface suitable for dyeing [26]. The yellowness index is inversely related to the whiteness, i.e., the higher the whiteness, the lower the yellowness index. The raw cotton yarn has the value of yellowness index (YI = 34.52) and the chemically bleached has the (YI = 1.75).

The raw cotton samples were also mercerized. Mercerization of cotton is performed in 20–30% NaOH solution with or without tension that achieves various effects such as increased moisture absorption, lustre and dyeability. The cotton fibre’s longitudinal view is converted from a ribbon shape to a straight shape, and the natural twist of cotton fibres disappears. Moreover, part of the crystalline region of the cotton fibres is converted into the amorphous region, so that the dyeability and chemical reactivity are improved. The mercerized cellulose has no chemical changes compared to the original cellulose, but the crystal structure is converted from cellulose I to cellulose II [27,31,32,33].

The samples in this experimental work have been mercerized by both methods, in slack (without the tension) and with tension. The results show the slight increase in whiteness compared to raw samples, although the mercerized samples were not scoured and bleached previously. This is due to the action of the sodium hydroxide, which removes the impurities that give the cotton a yellowish hue. The whiteness of sample mercerized in slack mercerization is W_CIE_ RSM = 5.60 and mercerization with tenacity is W_CIE_ RM4 = 1.40.

By processing pre-treated yarns with metal salts of copper, aluminium and ferrum (mordants) the whiteness and yellowness index also change. Mordants in addition to improving the binding of dyes to fibre, also have a major role in changing the whiteness, yellowness index and hue of undyed pre-treated yarns. Influence of pre-treatment with copper and aluminium, greatly affects the increase in whiteness compared to pre-treated yarns. All cotton yarns give a yellowish shade, except for chemically bleached copper-treated yarns. The hue of CB_Cu is in the green area (h° = 177.57) due to the natural greenish shade of CuSO_4_·5H_2_O solution. When cotton yarn was treated with ferrum mordant, the yellowness index increases relative to all pre-treatment yarns except raw yarn. The largest difference in yellowness index can be seen on RM4_Fe, which is YI = 38.54. Accordingly, the RM4_Fe yarn has the lowest lightness (L* = 79.92). Due to the removed pigments in chemically bleached yarns, the lightness is the highest and ranges from 93.49 to 90.92 (Table 3 and Table 4).

In the next step, the extraction of natural dyes from onion and pomegranate peel has been performed. As mentioned in the Introduction, one of the aims of this work was to investigate the possibility of using easily accessible natural sources (biowaste from food sources): pomegranate peel and onion peel for dyeing cotton yarn. The use of biowaste to obtain natural dyes is in line with one of the essential postulates of the circular economy, and that is waste minimization. It solves some of the basic problems for the possible commercial application of natural plant dyes: the need for plant sources, availability of raw materials, standardization of dyeing recipes, environmental and economic sustainability, i.e., the realization of the idea of zero greenhouse gas emissions. The idea of zero emissions is based on the idea that each biological waste can be a raw material for another production, i.e., that one industry can always consume the waste of another industry. Inexpensive by-products from agriculture and forestry can be used to obtain natural plant dyes, such as wood bark from the wood industry, industrial food and beverage waste such as pressed berries, distillation sludge and other residual by-products. Intensive use of industrial waste as renewable raw materials for the production of natural dyes would increase the economical use of waste material, contribute to the preservation of the environment and reduce the use of fossil fuels [34].

After the extraction, the absorption spectra of the extracts were measured spectrophotometrically. Figure 2 shows the absorption spectra of the extracts of onion and pomegranate peel in the ultraviolet and visible parts of the spectrum. In the visible part of the spectrum, the onion extract was confirmed by peaks at 250, 288 and 330 nm [35,36].

The pomegranate extract shows a maximum at 400 nm, i.e., in the yellow region, which is characteristic of punicalin. Punicalin in the aqueous extract of pomegranate was also confirmed by analysis in the ultraviolet part of the spectrum with peaks at 256, 278 and 360 nm [35,36].

Pomegranate peel (*Punica granatum* L.) contains 28% tannins. Pomegranate peel hydrolyzate can be gallic acid and ellagic acid as well as flavogalol. Tannins are punicalagin (2,3-(S)-hexahydroxydiphenoyl-4,6-(S,S)-galagyl-D-glucose; α-punicalagin; β-punicalagin) and punicalin (4,6-(S,S)-galagyl-D-glucose) and are responsible for obtaining yellow shades.

Onion peels (*Allium cepa* L.) are rich in flavonoids, and the most common is quercetin, followed by quercetin glucoside and kaempferol. Due to their chemical structure, flavonoids are the most common plant dyes of yellow shades. The natural plant dyes used in this paper belong to the group of acid-wetting dyes. Therefore, in the acid medium in dyeing process, ionization occurs on imide, carbonyl and hydroxyl groups, and depending on the raw material composition of the textile material, the choice of mordants (metal salts) and the chemical structure of the natural dye, metal complexes of different colours are formed. The obtained dyes and their properties are the result of the formation of a ligand: fibre-metal ion-natural dye (Figure 3) [34].

After dyeing, the samples were spectrophotometrically measured, and the analysis of objective colour parameters was performed. The results of the coordinate placement of the obtained colours are shown in a*/b* colour space, with a graphical representation of the objective values of the basic colour parameters: lightness (L*), chroma (C*) and hue (h°), in Figure 4.

From the colour coordinates of the cotton yarns dyed with onion peel (Figure 4), it is evident that the raw cotton yarn dyed in an aqueous extract of onion peel, achieved yellow hue (h° = 73.52) of high lightness (L* = 81.60) and low chroma (C* = 15.80). With the addition of mordants, the hue changes, but the most significant change occurred was the increase in chroma value. The sample treated with aluminium salts have chroma C* = 37.43 and with copper C* = 23.30. With lightness being L* = 68.81 and 63.67, respectively, the increase in colour intensity can be confirmed, although, given that it is a yellow hue, the ratio of chroma to lightness is still insufficient to give a clear, chromatic colour. As can be seen from the sample images (Table 4), the achieved shades are in the range of brownish-yellow-orange. The sample pre-treated with ferrum salts, as it was expected, achieved the most achromatic shade with the lowest chroma C* = 23.56.

Even more emphasized achromatic shade with lower chroma, is achieved for the scoured sample treated with ferrum salt (C* = 14.14), while for the rest of the scoured samples, the similar relationship of chroma C* and lightness L* have been achieved with hue ranging also in yellow-orange spectrum.

In chemically bleached cotton yarn, all natural pigments have been removed, so the substrate is whiter, giving the hues cleaner, lighter and more yellow after the dyeing. When mordant is added, the hue of the samples changes slightly (h° = 74.77–77.90) but it stays in yellow/yellow-orange spectrum. The sample treated with aluminium has the highest chroma value (C* = 33.53) which in relation to lightness (L* = 74.41) gives the highest colour intensity, although still in chromatic-achromatic area. For the scoured samples as well the most achromatic shade is achieved for the sample treated with ferrum salts.

In mercerized samples, a positive effect of mercerization on the absorbance and affinity of cotton fibre to dyes is observed. Even for mordant-free samples, a higher chroma value is achieved compared to non-mercerized samples, while for samples treated with mordants (even for samples treated with ferum salt) the highest intensity, arising from the specific lightness/chroma relationship is shown. Furthermore, the results show that for the samples treated with slack mercerization, a higher colour intensity is achieved compared to samples mercerized with tension.

The photographs of dyed samples regarding the cotton pre-treatments and mordant usage are shown in Table 5.

For samples dyed with natural dye extracted from pomegranate peel, the equal analysis is performed and the results of positioning the obtained colourations in a*/b* colour space with numerical evaluation of the main colour parameters; lightness (L*), chroma (C*) and hue (h°) is shown in Figure 5.

In general, a more pronounced yellow colour hue is achieved for samples dyed with pomegranate peel natural dye, compared to samples dyed with onion peel natural dye. Furthermore, higher chroma (C*) for non-mordanted samples is achieved compared to samples dyed with onion extracted dye. This arises from the fact that the pomegranate is a tannin dyestuff, and tannin has a property of behaving as a mordant. It is also important to mention that the pomegranate dye is a natural substantive dye which makes it suitable for cotton.

As for the samples treated with mordants, a similar influence on dye absorption and achieved relationship of chroma (C*) and lightness (L*) compared to the onion dyed samples can be seen. The sample treated with copper has the highest chromaticity (C* = 37.92), while the sample treated with ferrum salts has the lowest chroma value (C* = 13.67), which in relation to the lowest lightness (L* = 51.68) correspond to the visual appearance of the sample being dark greyish and the most achromatic.

The highest colour depth (which is inferred from the specific ratio of lightness and chroma) was obtained for the sample treated by slack mercerization and mordanted with copper salts.

The brightest shades are achieved for mercerized samples mordanted with aluminium salt. Although the value of chroma is not the highest for these samples, in a specific relationship with higher lightness, and given the nature of the yellow colour, visually, the appearance of a colour of greater brilliance is obtained (Table 6).

Contrary to theoretical expectations, the highest colour intensity was not obtained for bleached samples, although bleaching should improve colour clarity and brightness. Lower chroma relative to higher lightness resulted in a visually pastel, bright, unsaturated colour.

After colorimetric analysis of the dyed samples, a wash fastness test was performed. Five wash cycles were performed, and the results are shown after 1st and 5th cycles by the objective value of the total colour difference calculated according to the CIE76 system as well as by the value of grey scale grading (Table 7 and Table 8).

Samples treated with onion peel after the 1st and 5th wash cycle show low colour fastness, which was to be expected due to the chemical construction of the natural dye extracted from onion peel. After the 1st wash cycle, a smaller colour difference (DE) is visible than after the 5th wash cycle. The smallest colour difference is obtained for the sample pre-treated by slack mercerization, mordanted with CuSO_4_ (DE_RSM_O_Cu_ = 1.49). This is to be expected, because during free mercerization, the crystal lattice of cellulose changes. There is also an increase in amorphous areas in the cellulose that are responsible for higher dye binding. Grey scale values after 1st wash cycle, for most samples range between 3 and 4, which indicate rather satisfactory stability of natural dyes during washing, but after the 5th wash cycle, the wash fastness ratings decrease significantly, while DE increases. This indicates that during the 5th wash cycle, a significant amount of dye is removed from the yarns (Table 7).

Samples dyed with pomegranate peel showed better wash fastness compared to samples dyed with onion peel. Colour differences after the 1st wash cycle are in range from DE 1.42 (RSM_P_Fe) to 7.46 (RM4_P). After the 5th wash cycle, a minimal increase in total colour difference value (DE) is obtained, which indicate that in the 1st wash cycle, the unbounded dye was removed. The results of total colour difference (DE) obtained after the 5th wash cycles show that the optimal amount of dyestuff stayed bounded to the cotton fibre showing satisfactory wash fastness property. This is confirmed by the ratings in grey scale. The grades did not decrease drastically and did not decrease at all after the 5th wash cycle, which indicates satisfactory colour durability after the 5th wash cycle (Table 8).

Furthermore, the light fastness of obtained colorations was tested. Table 9 and Table 10 show the light fastness results of samples dyed with onion peel (Table 9) and pomegranate peel (Table 10) measured on Xenotest 440. The results are shown as objective numerical evaluation of total colour difference (DE) calculated according to CIE76 equation, as well as blue scale grades. The yarns dyed with pomegranate peel showed better light fastness. The colour difference (DE) after 41:10 h of light exposure of yarns dyed with pomegranate peel have ranges from 0.49 (R_P_Fe) to 7.35 (CB_P_Al), while yarns dyed with onion peel have higher colour differences ranging from 0.40 (RSM_O_Cu) to 18.81 (CB_O_Al). A significant influence of mordants on the results of light fastness was observed. Yarns dyed with onion mordanted with copper obtained better light fastness than other yarns. All yarns dyed with pomegranate peel also have the same trend. The yarns dyed with pomegranate peel mordanted with iron obtained better light fastness. An increase in colour difference (DE) was observed when using aluminium mordant for yarns dyed with both onion and pomegranate peel, which indicates a lower light fastness. The blue scale ratings confirm higher light fastness of yarns dyed with pomegranate peel dye, as well as a positive effect of mordanting.

## 4. Conclusions

Tests conducted in this paper confirm that, although natural dyes generally have a lower affinity for cellulose fibres, satisfactory dye quality results can be achieved. Furthermore, the positive effect of mercerization on the absorption and binding of dyes was confirmed, both in onions and pomegranates dyeing. Even without additional pre-treatment with mordants, certain colour intensity is obtained compared to untreated non-mercerized samples. However, in order to achieve more intense coloration, the necessity of applying mordants was confirmed. Optimal colour intensity results for the natural onion dye were obtained with aluminium salt as mordants, while for the pomegranate dye, the copper salts were confirmed as optimal.

Contrary to expectations, chemical bleaching did not improve the achievement of more intense coloration, but very light, pastel shades of low intensity were obtained.

Yarns treated with onion peel, after the 1st and 5th wash cycles show low colour fastness, which was to be expected due to the chemical constitution of the natural dye extracted from onion peel. For yarns dyed with pomegranate peel after the 5th wash cycle, a minimal increase in the total colour difference value (DE), compared to the 1st wash cycle was obtained, indicating a better wash fastness for yarns dyed with pomegranate peel.

As for the light fastness, for yarns dyed with pomegranate peel, better light fastness was obtained. It must be emphasized that the selection of mordant is also an important factor in light fastness. Thus, for the yarns dyed with onion peel mordanted with copper and yarns dyed with pomegranate peel mordanted with ferrum, satisfactory light fastness was obtained. For yarns mordanted with aluminium, the lowest light fastness on all dyed yarns were obtained.

## Figures and Tables

**Figure 1 molecules-27-04547-f001:**
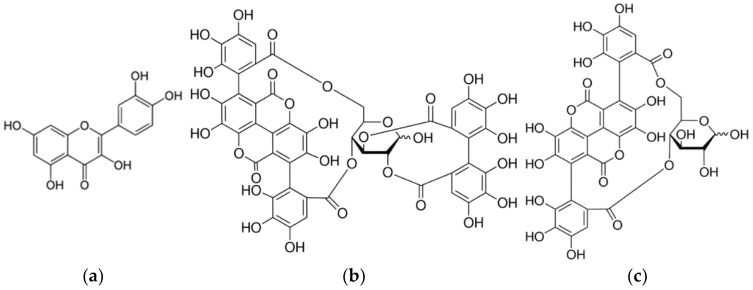
Chemical structures: (**a**) quercetin in onion peel and tannin derivatives in pomegranate peel; (**b**) punicagalin; (**c**) punicalin.

**Figure 2 molecules-27-04547-f002:**
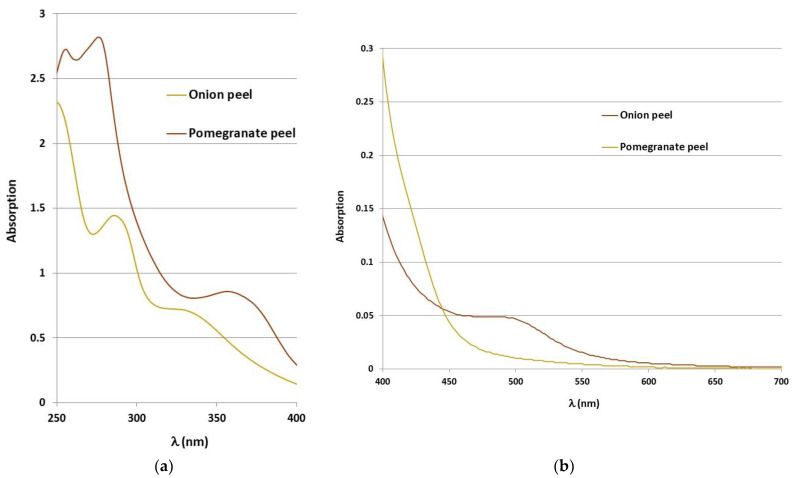
Absorption spectra of onion and pomegranate extracts in the (**a**) ultraviolet part and (**b**) visible part of the spectrum.

**Figure 3 molecules-27-04547-f003:**
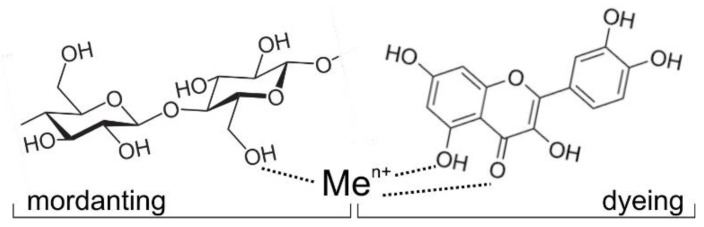
Schematic image of ligand formation: fibre—metal ion—natural dye.

**Figure 4 molecules-27-04547-f004:**
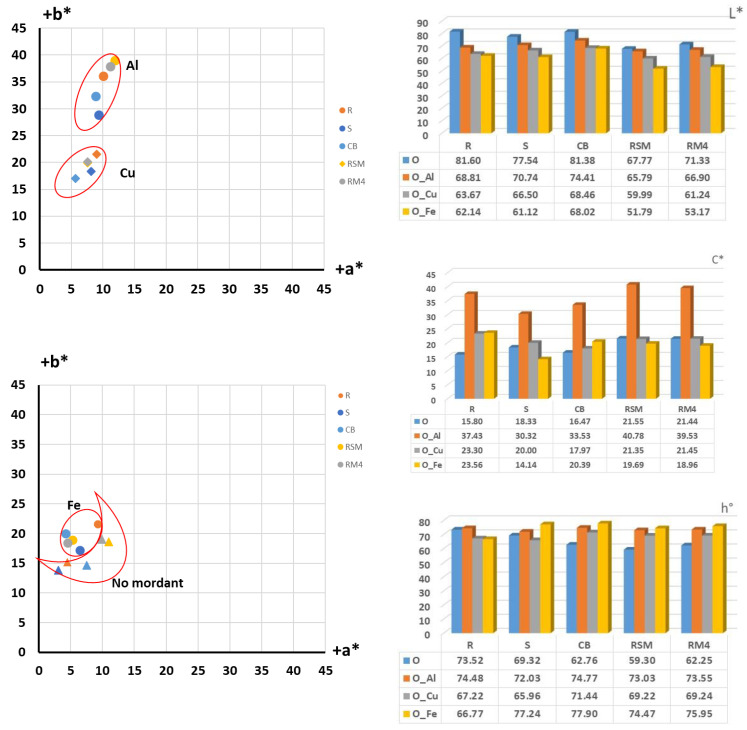
Colour analysis in the a*/b* colour space and objective results of lightness (L*), schroma (C*) and hue (h°) of cotton yarns dyed with onion peel.

**Figure 5 molecules-27-04547-f005:**
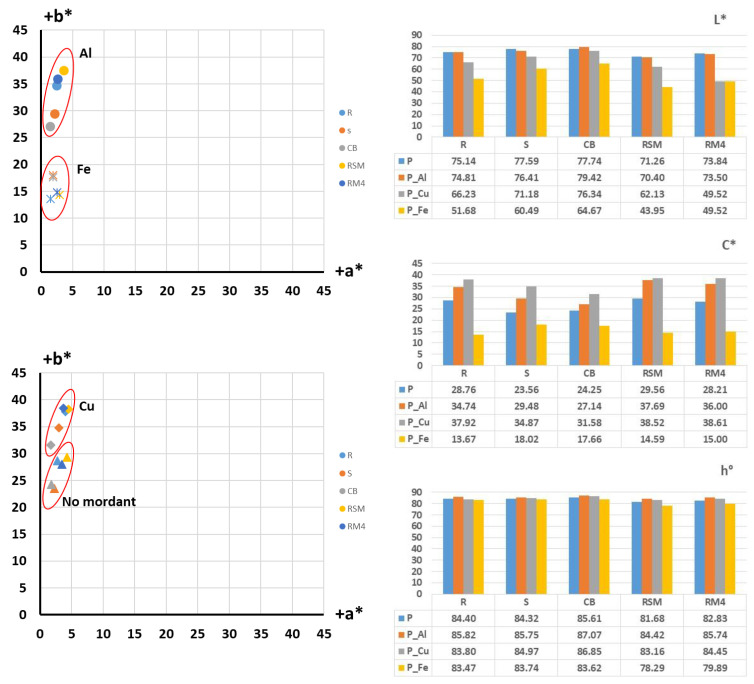
Colour analysis in the a*/b* colour space and objective results of lightness (L*), schroma (C*) and hue (h°) of cotton yarns dyed with pomegranate peel.

**Table 1 molecules-27-04547-t001:** Labels and treatments.

Labels	Treatments
R	Raw cotton yarn
S	Scoured cotton yarn
CB	Chemical bleach cotton yarn
RSM	Raw slack mercerized yarn
RM4	Raw mercerized yarn with 4% of tenacity
*O**	Cotton yarn dyed with onion peel
*P**	Cotton yarn dyed with pomegranate peel
*Al	Cotton yarn mordanted with: KAl(SO_4_)_2_·12H_2_O
*Cu	Cotton yarn mordanted with: CuSO_4_·5H_2_O
*Fe	Cotton yarn mordanted with FeSO_4_·7H_2_O

*—Raw cotton yarn (R), Scoured cotton yarn (S), Chemical bleach cotton yarn (CB), Raw slack mercerized yarn (RSM), Raw mercerized yarn with 4% of tenacity (RM4) dyed with onion peel (O) or pomegranate peel (P). **—Raw cotton yarn (R), Scoured cotton yarn (S), Chemical bleach cotton yarn (CB), Raw slack mercerized yarn (RSM), Raw mercerized yarn with 4% of tenacity (RM4) dyed with onion peel (O) or pomegranate peel (P) without or with mordanted (KAl(SO_4_)_2_·12H_2_O, CuSO_4_·5H_2_O, FeSO_4_·7H_2_O).

**Table 2 molecules-27-04547-t002:** Pre-treatment procedures.

Pre-Treatment	Scouring	Chemical Bleaching	Mercerization (Slack/with Tenacity 4%)
**Batch Composition**	NaOH (3%) Felosan (CHT Group): 2g/L	H_2_O_2_ (35%): 25 mL/L NaOH: 4 g/L Tinoclarit CBB (Ciba, Swiss): 5 mL/L Water glass (mixture of Na_2_SiO_3_, Na_2_Si_2_O_5_): 15 mL/L Heptol ESW (CHT Group): 10 mL/L Fumexol DF (Ciba, Swiss): 0.5 mL/L	NaOH (24%) Subitol MLF (Bezema): 8 g/L
**Methods**	Process of exhaustion	Process of exhaustion	Slack mercerization: laboratory beaker Mercerization with tenacity 4%: mercerizing machine
**Proces Parameters**	T = 100 °C t = 30 min	T = 100 °C t = 30 min	T = 16 °C t = 120 s
**Bath Ratio**	1:20	1:20	/
**After Pretreatmant**	Rinsing (hot soft water, medium hot water and cold soft water)	Rinsing (hot soft water, medium hot water and cold soft water) Neutralization (1% CH_3_COOH) Rinsing (soft water until neutral)	Rinsing (hot soft water, medium hot water and cold soft water) Neutralization (1% CH_3_COOH) Rinsing (soft water until neutral)

**Table 3 molecules-27-04547-t003:** Whiteness, yellowness index and spectral characteristics of pre-treatment and mordanted yarns.

Labels	W_CIE_	YI	L*	a*	b*	C*	h°
R	−17.00	34.52	84.42	2.92	16.48	16.74	79.95
R_Cu	−3.10	25.18	82.68	−0.40	12.64	12.65	91.80
R_Al	−3.00	29.03	84.60	2.28	13.74	13.93	80.58
R_Fe	−19.20	32.49	81.03	2.28	15.08	15.25	81.42
S	40.10	14.19	88.03	1.17	6.64	6.74	79.98
S_Cu	39.20	13.59	88.38	−0.28	6.97	6.98	92.31
S_Al	47.10	12.92	90.16	0.85	6.25	6.30	82.25
S_Fe	−0.50	28.88	85.54	2.42	13.73	13.94	80.01
CB	79.70	1.75	93.49	−0.20	0.96	0.98	101.94
CB_Cu	78.10	−0.86	90.98	−1.25	0.05	1.25	177.57
CB_Al	85.40	1.56	95.71	−0.18	0.87	0.89	101.97
CB_Fe	21.30	23.71	90.92	1.12	13.13	12.18	84.73
RSM	5.60	24.31	83.44	1.52	11.38	11.48	82.41
RSM _Cu	−1.50	21.24	79.86	−1.95	10.96	11.13	100.8
RSM_Al	14.00	22.51	85.99	0.97	10.96	11.01	84.96
RSM_Fe	−8.30	29.68	83.21	1.95	14.04	14.18	82.09
RM4	1.40	24.66	81.85	1.38	11.44	11.52	83.13
RM4_Cu	11.00	18.77	82.37	−1.48	9.73	9.84	98.62
RM4_Al	6.20	23.67	83.20	1.23	11.14	11.21	83.69
RM4_Fe	−35.50	38.54	79.92	3.37	17.64	17.96	79.17

**Table 4 molecules-27-04547-t004:** Pre-treated and mordanted cotton yarns.

Treatments	Mordants			
	-	Al	Cu	Fe
R	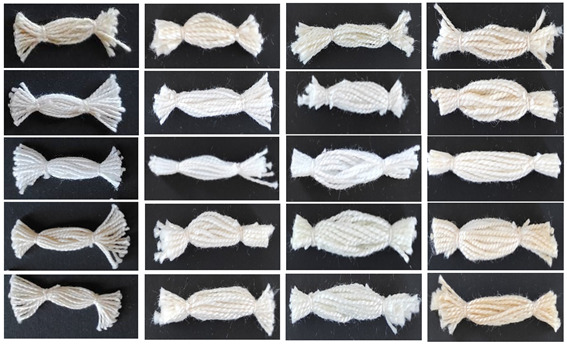			
S				
CB				
RSM				
RM4				

**Table 5 molecules-27-04547-t005:** Cotton yarns dyed with onion peel.

Treatments	Mordants			
	-	Al	Cu	Fe
R_O	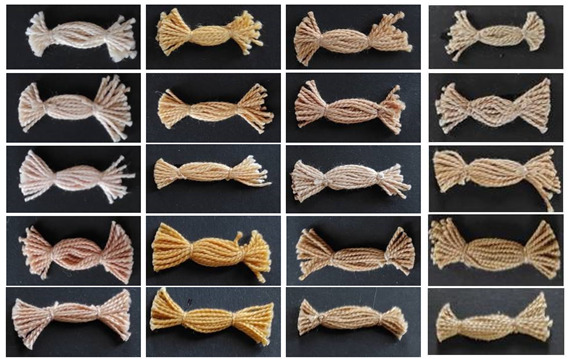			
S_O				
CB_O				
RSM_O				
RM4_O				

**Table 6 molecules-27-04547-t006:** Cotton yarns dyed with pomegranate peel.

Treatments	Mordants			
	-	Al	Cu	Fe
R_P	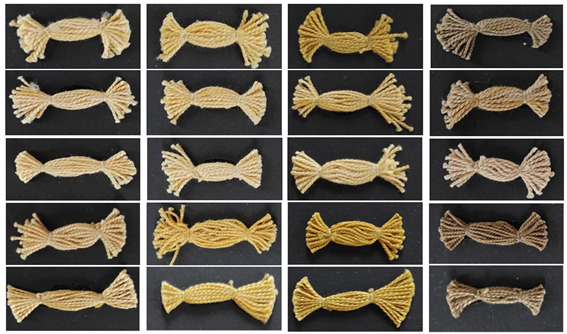			
S_P				
CB_P				
RSM_P				
RM4_P				

**Table 7 molecules-27-04547-t007:** Colour fastness to washing after 1st and 5th cycles of washing of cotton yarns dyed with onion peel.

(a) Objective evaluation	(b) Grey scale evaluation		
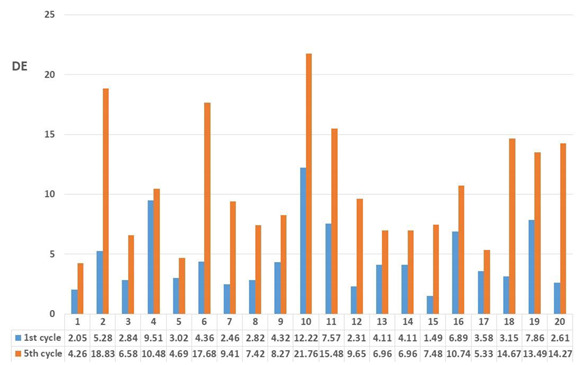	Labels	1st	2nd
	1	4	3
	2	3	1
	3	3	2–3
	4	3–4	2
	5	3–4	2–3
	6	3	1
	7	3–4	2
	8	3	2
	9	3	2
	10	1–2	1
	11	2	1
	12	3–4	1–2
	13	3	2
	14	3	1–2
	15	4	2
	16	2	1–2
1 (R_O); 2 (R_O_Al); 3 (R_O_Cu); 4 (R_O_Fe); 5 (S_O); 6 (S_O_Al); 7 (S_O_Cu); 8 (S_O_Fe); 9 (CB_O); 10 (CB_O_Al); 11 (CB_O_Cu); 12 (CB_O_Fe); 13 (RSM_O); 14 (RSM_O_Al); 15 (RSM_O_Cu); 16 (RSM_O_Fe); 17 (RM4_O); 18 (RM4_O_Al); 19 (RM4_O_Cu); 20 (RM4_O_Fe)	17	3	2–3
	18	3–4	1–2
	19	2	1
	20	3–4	1

**Table 8 molecules-27-04547-t008:** Colour fastness to washing after 1st and 5th cycles of washing of cotton yarns dyed with pomegranate peel.

(a) Objective evaluation	(b) Grey scale evaluation		
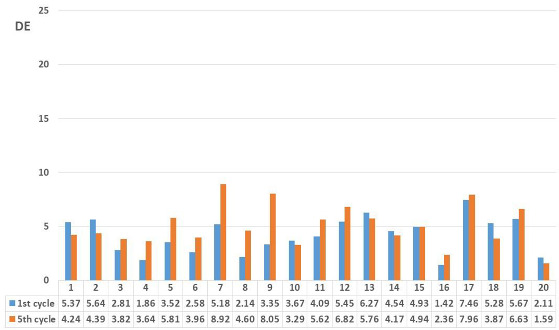	Labels	1st	2nd
	1	3	3
	2	2–3	3
	3	3–4	3
	4	4	3
	5	3	2
	6	3	3
	7	2	2
	8	4	2–3
	9	3	2
	10	3	3
	11	3	3
	12	2	2
	13	2–3	2–3
	14	3	3
	15	3	3
	16	4	3–4
1 (R_O); 2 (R_O_Al); 3 (R_O_Cu); 4 (R_O_Fe); 5 (S_O); 6 (S_O_Al); 7 (S_O_Cu); 8 (S_O_Fe); 9 (CB_O); 10 (CB_O_Al); 11 (CB_O_Cu); 12 (CB_O_Fe); 13 (RSM_O); 14 (RSM_O_Al); 15 (RSM_O_Cu); 16 (RSM_O_Fe); 17 (RM4_O); 18 (RM4_O_Al); 19 (RM4_O_Cu); 20 (RM4_O_Fe)	17	2	3
	18	2–3	3
	19	2–3	2–3
	20	4–3	4

**Table 9 molecules-27-04547-t009:** Light fastness of cotton yarns dyed with onion peel.

(a) Objective evaluation	(b) Blue scale evaluation	
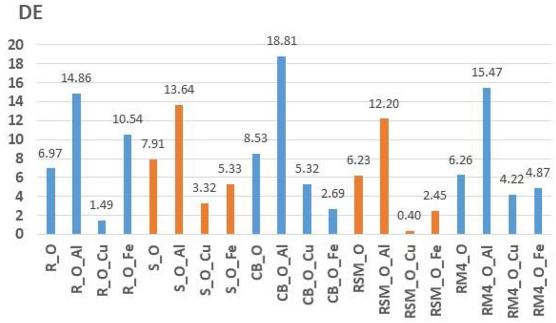	Labels	1st
	R_O	4–5
	R_O_Al	4
	R_O_Cu	6
	R_O_Fe	4
	S_O	4–5
	S_O_Al	4
	S_O_Cu	5
	S_O_Fe	5
	CB_O	4–5
	CB_O_Al	3
	CB_O_Cu	5
	CB_O_Fe	5
	RSM_O	5
	RSM_O_Al	4
	RSM_O_Cu	8
	RSM_O_Fe	6
	RM4_O	5
	RM4_O_Al	4
	RM4_O_Cu	5
	RM4_O_Fe	5

**Table 10 molecules-27-04547-t010:** Light fastness of cotton yarns dyed with onion peel.

(a) Objective evaluation	(b) Blue scale evaluation	
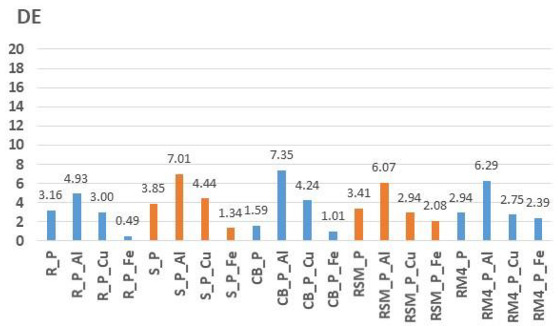	Labels	1st
	R_O	5
	R_O_Al	5
	R_O_Cu	5
	R_O_Fe	8
	S_O	5
	S_O_Al	4–5
	S_O_Cu	5
	S_O_Fe	6
	CB_O	6
	CB_O_Al	4–5
	CB_O_Cu	5
	CB_O_Fe	7
	RSM_O	5
	RSM_O_Al	5
	RSM_O_Cu	6
	RSM_O_Fe	5
	RM4_O	5
	RM4_O_Al	5
	RM4_O_Cu	5
	RM4_O_Fe	5–6

## Data Availability

The data presented in this study are available on request from the corresponding author.

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
