# Peer review of "Influence of Cotton Pre-Treatment on Dyeing with Onion and Pomegranate Peel Extracts"

_molecules, 2022, doi:10.3390/molecules27144547_

Round 1
Reviewer 1 Report
line 11-12 in Abstract - Latin names should be written in italics (Allium cepa L.) (Punica granatum L.).
The authors should improve the introduction more carefully, in my opinion lines 29-33 and 33-37 are almost the same text. Needs the necessary correction!
I believe in deep knowledge and awareness of potential readers but it would be worthwhile to initially alternate UV with fading. You can also add such an explanation of the sentence. In my opinion, the purpose of the work is not clearly formulated, I have read it from the context of course, but it should be clearly marked. The current text is more indicative of what has been done, not what and what research has been undertaken.
2.1. Pre-treatmants of cotton yarns - in this chapter, it is unnecessary to refer to table 2 twice
table 2 requires more attention when formatting because the lines related to the method in the first column are not properly assigned to the lines from the last column
I think that all chemical formulas also need to be corrected so that the number of atoms of a given first grid in a molecule / compound is a digit with a subscript rather than a letter diminished
the abbreviations referring to the square meter (line 207-208 and probably the whole work) were also formatted with a little care, it should be a superscript
table 3 requires the abbreviations used in it to be added. Besides, it is assumed that the measurements were made in at least 3 repetitions, it may be worth adding an element of a very basic statistical analysis - such as standard deviation
table 4-5 requires the abbreviations used in it to be added.
Reviewer 2 Report
This work proved the possibility of cotton material dyeing with natural dyes extracted from onion and pomegranate peel. I think this manuscript is well structured and I believe it will provide new insights into the development of the complex problem of dyeing cellulose fibres with natural dyes and the circular economy. The article is solid and the information is comprehensive. Therefore, I recommend accept it for publication in this journal after some revision.
(1)The author needs to further condense the summary and conclusion to make it easier for readers to understand.
(2)The author should provide some water resistance results to show the stability of the system.
(3)I suggest authors should be compared with other references to highlight the advantages of this method in this paper.
Reviewer 3 Report
Authors implemented the research from the extraction of natural dyes to their application on cotton, which is comprehensive and informative. The topic is also an appealing one from the perspective of sustainable textile industry. Although authors have tried to revise the paper, the MS overall appears more like a technical report rather than a scientific research. The discussions are a bit superficial and the overall MS is not focused on solving the core problem of natural dyeing.
Importantly, there are already similar works that have been published (See list below). How this work makes a step forward upon or distinguishes from these studies should be considered.
1. Davulcu, A., Benli, H., Şen, Y. et al. Dyeing of cotton with thyme and pomegranate peel. Cellulose 21, 4671–4680 (2014). https://doi.org/10.1007/s10570-014-0427-8
2. Cotton Dyeing with Natural Dye Extracted from Pomegranate (Punica granatum) Peel. Source: Universal Journal of Environmental Research & Technology . Aug2011, Vol. 1 Issue 2, p135-139. 5p.
3. Satyanarayana, D. N. V., & Chandra, K. R. (2013). Dyeing of cotton cloth with natural dye extracted from pomegranate peel and its fastness. International Journal of Engineering Sciences & Research Technology, 2(10), 2664-69.
4. Satyanarayana, D. N. V., & Chandra, K. R. (2013). Dyeing of cotton cloth with natural dye extracted from pomegranate peel and its fastness. International Journal of Engineering Sciences & Research Technology, 2(10), 2664-69.
5. Adeel, S., Ali, S., Bhatti, I. A., & Zsila, F. (2009). Dyeing of cotton fabric using pomegranate (Punica granatum) aqueous extract. Asian Journal of Chemistry, 21(5), 3493.
6. M G Silva et al 2018 IOP Conf. Ser.: Mater. Sci. Eng. 460 012032 Multifunctionalization of cotton with onion skin extract.
7. Gesese, T. N., Fanta, S. W., & Mersha, D. A. (2020, October). Antimicrobial Activity of Cotton Fabric Treated with Solanum Incanum Fruit and Red Onion Peel Extract. In International Conference on Advances of Science and Technology (pp. 68-82). Springer, Cham.
8. …
Thus, I do not recommend it for publish. If the authors could address one or two critical issues (e.g. washing/light fastness of natural dyeings, affinity of natural extract to cellulose, heavy colour change after mordanting) during the natural dyeing of cotton after careful revision, then I would like to reconsider it for publication.
Round 2
Reviewer 3 Report
Authors have tried their best to improve the MS, which is sufficient to be published as it is.